

# TransPrise: a novel machine learning approach for eukaryotic promoter prediction

Stepan Pachganov[1,*], Khalimat Murtazalieva[2,3,*], Aleksei Zarubin[4], Dmitry Sokolov[5], Duane R. Chartier[6] and Tatiana V. Tatarinova[2,7,8,9]

[1] Ugra Research Institute of Information Technologies, Khanty-Mansiysk, Russia
[2] Vavilov Institute for General Genetics, Moscow, Russia
[3] Institute of Bioinformatics, Moscow, Russia
[4] Tomsk National Research Medical Center of the Russian Academy of Sciences, Research Institute of Medical Genetics, Tomsk, Russia
[5] Neirika Solutions, Sarov, Russia
[6] International Center for Art Intelligence, Inc., Los Angeles, CA, United States of America
[7] Department of Biology, University of La Verne, La Verne, CA, United States of America
[8] A.A. Kharkevich Institute for Information Transmission Problems, Russian Academy of Sciences, Moscow, Russia
[9] Siberian Federal University, Krasnoyarsk, Russia
[*] These authors contributed equally to this work.

Corresponding author
Tatiana V. Tatarinova,
ttatarinova@laverne.edu

## ABSTRACT

As interest in genetic resequencing increases, so does the need for effective mathematical, computational, and statistical approaches. One of the difficult problems in genome annotation is determination of precise positions of transcription start sites. In this paper we present TransPrise—an efficient deep learning tool for prediction of positions of eukaryotic transcription start sites. Our pipeline consists of two parts: the binary classifier operates the first, and if a sequence is classified as TSS-containing the regression step follows, where the precise location of TSS is being identified. TransPrise offers significant improvement over existing promoter-prediction methods. To illustrate this, we compared predictions of TransPrise classification and regression models with the TSSPlant approach for the well annotated genome of *Oryza sativa*. Using a computer equipped with a graphics processing unit, the run time of TransPrise is 250 minutes on a genome of 374 Mb long. The Matthews correlation coefficient value for TransPrise is 0.79, more than two times larger than the 0.31 for TSSPlant classification models. This represents a high level of prediction accuracy. Additionally, the mean absolute error for the regression model is 29.19 nt, allowing for accurate prediction of TSS location. TransPrise was also tested in *Homo sapiens*, where mean absolute error of the regression model was 47.986 nt. We provide the full basis for the comparison and encourage users to freely access a set of our computational tools to facilitate and streamline their own analyses. The ready-to-use Docker image with all necessary packages, models, code as well as the source code of the TransPrise algorithm are available at (http://compubioverne.group/). The source code is ready to use and customizable to predict TSS in any eukaryotic organism.

## INTRODUCTION

The heightened interest and lowered costs of sequencing has led to intensified use of mathematical and statistical tools to improve the accuracy and replicability of genomic analysis. Until recently, there has been a relatively simplistic and deterministic notion of promoter organization and the creation of functional models of gene regulation. We are at a turning point in genetic research—we are actively assembling and refining the "dictionary" of genetics, but we are in the infancy of development of "genetic grammar". Multiple questions remain to be addressed: what do the individual "words" mean? How does genotype translate to phenotype? This is very much like the time in physics, over 100 years ago, when the classical approach had to be replaced by quantum theory and statistical mechanics. We cannot understand genetics without powerful statistical and computational tools for analysis and for verifying our hypotheses regarding genome manifestations.

Thousands of eukaryotic genomes have been sequenced so far (https://www.ncbi.nlm.nih.gov/genome/browse/#!/eukaryotes/), including animals (1,590), fungi (3,275), and plants (665). As of 2018, these genomes are at various assembly levels, with 840 genomes assembled at the level of chromosomes, 46 are complete, 1,191 are in contigs, and 4,057 are at the level of genomic scaffolds. Genomic projects are not limited to sequencing and genome assembly. Re-sequencing large populations is becoming an important tool to unravel population structure, detect signatures of selection and to map quantitative trait loci (QTL) (*Atwell et al., 2010*). As resequencing costs plummet and technology platforms continue to expand throughput (e.g., Illumina NovoSeq), genomics communities are now contemplating the possibilities of resequencing entire germplasm collections to detect the vast majority of existing alleles and haplotypes. One essential requirement to capture allelic diversity is to have high-quality reference genomes that span the breadth of genomic diversity for mapping resequencing data.

Understanding the functional role of a given single-nucleotide polymorphism or a structural variant requires knowledge of its location with respect to coding and regulatory regions and the elements involved (*Li et al., 2015*; *Gao et al., 2018*; *Tatarinova et al., 2016*; *Triska et al., 2017b*). In addition, the regulatory role of a transcription factor binding site (TBFS) has been demonstrated to depend on the position of the TFBS with respect to the transcription start site (TSS) (*Berendzen et al., 2006*; *Pritsker et al., 2004*). Determination of the precise location of TSS is an essential preparatory step for motif discovery and reconstruction of gene regulatory networks (*Triska et al., 2017a*; *Troukhan et al., 2009*). The interaction of a vast number of proteins, multi-subunit complexes, and DNA binding sites make eukaryotic transcriptional regulation an extremely convoluted process (*Eckardt, 2014*). Therefore, it is vitally important to have reliable methods for promoter prediction

and analysis of regulatory elements if we are to enhance our capacity to engineer crops or to select therapeutic targets.

Homology-based prediction of coding regions is a relatively straightforward procedure (*Keilwagen et al., 2018*). Multiple tools and pipelines exist for finding positions and functions of genes, such as MAKER (*Campbell et al., 2014a*; *Campbell et al., 2014b*; *Holt & Yandell, 2011*), BREAKER (*Hoff et al., 2015*), Augustus (*Stanke & Morgenstern, 2005*), GeneMarkHMM (*Lukashin, 1998*), FgeneSH (*Salamov & Solovyev, 2000*), and many others. These pipelines achieve remarkably high accuracy in homology-based gene finding; however, homology between species does not necessarily extend beyond coding regions, and, therefore, accurate prediction of promoters is difficult. It has been reported that even state-of-the art modern methods of promoter mapping are incapable of achieving 100% accuracy (*Alexandrov et al., 2009*; *Alexandrov et al., 2006*; *Batut et al., 2013*; *Carninci et al., 2006*; *Herbig, Sharma & Nieselt, 2013*; *Kawaji et al., 2006*; *Kawaji et al., 2014*; *Morton et al., 2014*; *Tatarinova et al., 2013*; *Tatarinova et al., 2016*; *Troukhan et al., 2009*). For example, current annotations of rice (MSU7) and maize (B73, 6a) contain 56K and 63K predicted genes, correspondingly (*Liseron-Monfils et al., 2013*), and for nearly two–thirds of those genes, TSS is not identified precisely. Traditional deterministic approaches can predict only ∼50% of promoters with one false positive promoter predicted every 700–1,000 nt of the genome (*Shahmuradov & Solovyev, 2015*; *Solovyev, Shahmuradov & Salamov, 2010*). This accuracy is insufficient to make reliable predictions, because we expect one promoter per 10,000–20,000 nt of a genome. PromH (*Solovyev & Shahmuradov, 2003*) used conservation of promoter functional components between orthologous genes to improve prediction of TSS. PromH was able to predict TSS within 10 nt for 90% of the TATA+ promoters and for 40% of TATA- genes, but only if there are highly similar homologous sequences from closely related species. The TSSer algorithm (*Troukhan et al., 2009*) that combined positional frequency of 5′ EST/RNA-Seq matches on genomic DNA with gene models was able to accurately predict one transcription start site per locus. However, it is now accepted that alternative promoters are associated with differential expression in various tissues and chromatin states (*Rye et al., 2014*). A nonparametric maximum likelihood approach, NPEST (*Tatarinova et al., 2013*), can predict multiple TSSs per locus if 5′ EST/CAGE/mRNA data are available. Promoter sequences predicted by NPEST were demonstrated to be more accurate for the *A. thaliana* genome than sequences identified in several gold standard databases, such as TAIR, Plant Prom DB and Plant Promoter Database. However, it is difficult to identify TSS from RNA-Seq alone, since only 26% of genes display a maximum of the RNA-Seq coverage in the range [TSS-50, TSS + 250], and only 60% of genes display this maximum in the range [TSS-50, TSS + 550] (*Steijger et al., 2013*). Enough RNA-Seq and CAGE data is not available for all genomes of interest. Therefore, it is imperative to develop alternative strategies.

There are several factors complicating the process of TSS prediction, such as existence of multiple TSS per locus. Studies on mammalian and plant genomes have revealed that many eukaryotic genes are associated with multiple distinct promoters (*Batut et al., 2013*; *Farrell & Bassett, 2007*; *Louzada, 2007*; *Morton et al., 2014*; *Tatarinova et al., 2013*). Moreover, eukaryotic promoters are characterized by multiple TSSs and can be classified based on

the distribution and utilization of their collective TSSs. Consequently, the association with several distinct promoters allows for a single gene to encode various protein isoforms (*Sandelin et al., 2007*).

In addition, the performance of standard promoter identification in grasses and warm-blooded vertebrates is complicated by the existence of two classes of genes in those organisms: $GC_3$—rich and—poor ones (where $GC_3$ is the fraction of Cs and Gs in the third position of codons). Nucleotide composition of $GC_3$—rich genes differs from $GC_3$—poor ones; they also have higher variability of gene expression levels (resulting in fewer full-length mRNA support) (*Elhaik, Pellegrini & Tatarinova, 2014*; *Elhaik & Tatarinova, 2012*; *Tatarinova, Elhaik & Pellegrini, 2013*). Since a majority of the stress-related and tissue-specific genes are $GC_3$-rich (*Chan et al., 2017b*), refinement of promoter prediction pipeline is an essential task.

Many genomic features are associated with the location of promoter: positional frequency of 5′ ESTs and RNA-Seq matches on genomic DNA, nucleotide distribution, DNA methylation, distribution of SNPs, and characteristic regulatory elements. Incorporation of those data types allows accurate prediction of TSS. A recently developed tool, TSSPlant (*Shahmuradov, Umarov & Solovyev, 2017*), based on the Expectation Maximization (EM) algorithm, achieves significantly higher accuracy compared to state-of-the art promoter prediction programs for both TATA-containing and TATA-less promoters. *Umarov & Solovyev (2017)* developed a deep learning approach to characterize genomic regions as promoters and non-promoters; and *Triska et al. (2017b)* applied it to the rice genome, achieving 99% accuracy in classification of 250 nt long regions. However, the question of the specific location of the TSS within these 250 nt long windows remains open.

This paper presents a novel, accurate, and data-type independent procedure for TSS prediction that can incorporate multiple data types. Our method is based on a machine learning approach that is capable of uncovering intricate properties of promoter regions and achieving much higher accuracy than deterministic methods (*Umarov & Solovyev, 2017*). Our novel method aims to identify the position of the start of transcription with the highest possible precision using nucleotide composition alone. The method can predict multiple transcription start sites per locus. It is data-type agnostic and can be extended to incorporate additional biological features. We present a set of computational tools, a user-friendly public interface and a curated online database to facilitate these analyses.

## MATERIALS AND METHODS

### Selection of genome annotation version

We selected rice chromosomes and Genome Annotation release 7 (MSUv7, http://rice.plantbiology.msu.edu). There are two commonly used annotations of rice: MSU (*Kawahara et al., 2013*) and FgeneSH (*Zhang et al., 2008*). The Fgenesh gene prediction set contains 18,389 high quality (5′ full, with mRNA support) gene models, while the MSU gene prediction set contains 20,367 high quality gene models (*Tatarinova et al., 2016*). We used Fgenesh mRNA-based gene prediction models, since Fgenesh-annotated promoters have a more pronounced nucleotide consensus as compared to the promoters annotated by

MSU (*Triska et al., 2017b*). Fgenesh was successfully used to annotate several plant genomes (*Chan et al., 2017a*; *Chan et al., 2017b*; *Davis et al., 2010*; *Ito et al., 2005*; *Jiang et al., 2015*; *Nasiri et al., 2013*; *Sanusi et al., 2018*; *Sheshadri et al., 2018*; *Yao et al., 2005*). Therefore, we selected the Fgenesh annotation as the gold standard for our analysis. To obtain the highest quality dataset, pseudogenes, transposable elements, and genes with 5′ UTR shorter than 20 nt or longer than 1,000 nt have been excluded.

## Training, validation and test sets

The procedure consists of two steps: classification (dividing the genome into "promoters" and "non-promoters") and regression (finding the position of TSS inside the sequence identified as "promoter"). The genomic sequence of *Oryza sativa* had been divided into the testing and training sets. The training set contains three files:

1. *Training "non-promoter" dataset* contains sequences extracted from random genomic positions separated from experimentally validated transcription start sites by 2,000 nt. This dataset contains mostly intergenic regions. All sequences are 2,000 nt long.

2. *Training "promoter" dataset* contains sequences [TSS-1,000; TSS+999] from the all chromosomes with length 2,000 nt.

3. File with *indicators of TSS positions*, containing $(2,000 \times 1)$ matrices that correspond to positions of biologically validated TSS in every training sequence ("1" TSS, "0" not TSS position).

The same set of files was created for the testing dataset. Since the procedure has multiple steps (classification and regression), training and testing sets were selected at each step of the method.

The following procedure was used to assemble the dataset for the **classification model**:

(1) "Non-promoters": $\frac{1}{4}$ of the examples chosen from the training "non-promoter" dataset, randomly selecting 512 nt long sequences from 2,000 nt long regions.

(2) "Promoters sans TSS": $\frac{1}{4}$ of the examples were randomly selected from the training "promoter" dataset, making sure that the chosen 512 nt long fragment did not overlap the region [TSS-50, TSS+50].

(3) "TSS vicinity": $\frac{1}{2}$ of the examples extracted from the training "promoter" dataset, containing only one TSS in a random position within the 512 nt long sequence, with a restriction that it should be in the [250, 450] fragment.

The dataset for the **regression model** was assembled using sequences that contain one validated TSSs in a randomly selected position of the [250, 450] fragment. The datasets are represented as $(512 \times 4)$ nucleotide matrices $M$ with 512 columns and 4 rows. The 1st row contains delta function $\delta(x_i = A)$—it is equal to 1 if there is nucleotide "A" in the $i$th position of the sequence and 0 otherwise. Similarly, 2nd, 3rd, and 4th rows correspond to nucleotides C, G and T.

## Model training

We implemented the Convolutional Neural Networks (CNN) using the Keras library for training (https://keras.io/).

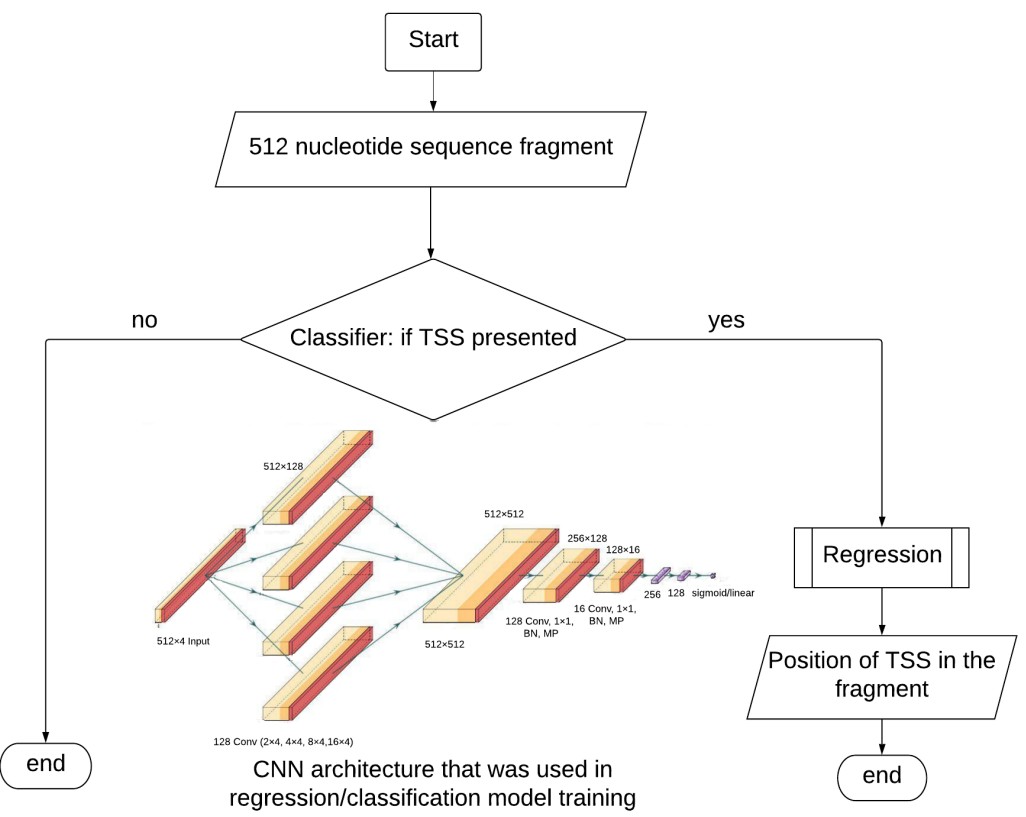

Figure 1 TransPrise algorithm. Inset: CNN architecture that was used in classification/regression model training. CNN architecture was used in classification and regression steps. BN, batch normalization; MP, max. pooling.

## Classification and Regression models training

The dataset for the **classification** contains equal numbers of positive and negative examples. The matrices (512 × 4) described above are input into the model. The CNN architecture (Fig. 1) started with four parallel convolutional layers (composed of 128 filters with 2, 4, 8 and 16 kernel sizes) ReLU, was used as activation function followed by concatenation. After concatenation layer we used convolution, batch normalization, max pooling layers twice. The first convolution had 128 filters and the second had 16. There was one kernel size and ReLU activation in both situations. To help regularize the model, we used the 0.5 Dropout technique. The signal is fed to two fully connected layers with ReLU activation functions consisting of 256 and 128 neurons, followed by batch normalization. The output layer had a sigmoid activation function.

## Whole Genome Sequencing (WGS) processing

TransPrise also works with WGS data in *fasta* format. We prepared a Python script that with 250 min run time on a genome of 374 Mb long. It uses a sliding window for extraction of 512 nt long sequences with the step size 4 nt. If the classification step identifies a fragment as TSS-containing, this fragment will be passed to the regression step. The prediction
vector is compared to the TSS model using a L1 norm. The default value for the similarity threshold is 600, but it can be modified by users. The only restriction of our algorithm is that alternative TSSs must be more than 100 nt of each other.

## Model evaluation

For the purpose of the K-fold cross-validation, the dataset is randomly divided into K equal-size subsets. Of the K subsets, a single subset is retained as the validation data for testing the model, and the remaining (K-1) subsets are used for training. The cross-validation process is then repeated K times (the ''folds''), with each of the K subsamples used exactly once as the validation data. Then, the results from K folds are averaged. The advantage of K-fold cross-validation is that all observations are used for both training and validation, and each observation is used for validation exactly once. We conducted 10-fold cross-validation (dataset was divided into training and validation sets in 9:1 ratio; validation set was used to avoid overfitting and find the optimal number of learning epochs). The ROC curves obtained in 10-fold cross-validation are presented in the ''Results'' section. We determined that the optimal number of learning epochs is five. After the model training, we tested our model using the test set and calculated Accuracy (Ac), Sensitivity (Se), Specificity (Sp), and the Matthews Correlation Coefficient (CC):

$$\text{Specificity} = \frac{TP}{TP + FP},$$

$$\text{Sensitivity} = \frac{TP}{TP + FN},$$

$$\text{Accuracy} = \frac{TP + TN}{TP + TN + FP + FN},$$

$$CC = \frac{TP \times TN + FP \times FN}{\sqrt{(TP + FP)(TP + FN)(TN + FP)(TN + FN)}},$$

where TP—true positive, TN—true negative, FP—false positive, FN—false negative. The input of the regression model has the same format as the classification model input. TSS is assumed to be located at a random position between nucleotides 250 and 450. There is only one difference between classification and regression models: in the output layer the activation function is replaced by a linear function.

We trained the classification and regression models using the Keras Library and performed 10-fold cross validation procedures for them. The algorithm for TSS prediction is presented in Fig. 1. For every fold, we carried out five learning epochs. The complete learning time was 35 s on average. We performed 10-fold cross-validation and calculated the average value of mean absolute error (MAE) to estimate the accuracy of TSS position prediction, where $y_i$—position of TSS in test set (assumed to be accurate), and $x_i$—predicted position of TSS.

$$\text{MAE} = \frac{\sum_{i=1}^{n} |y_i - x_i|}{n}.$$

Figure 2 shows the flowchart for model building and validation.

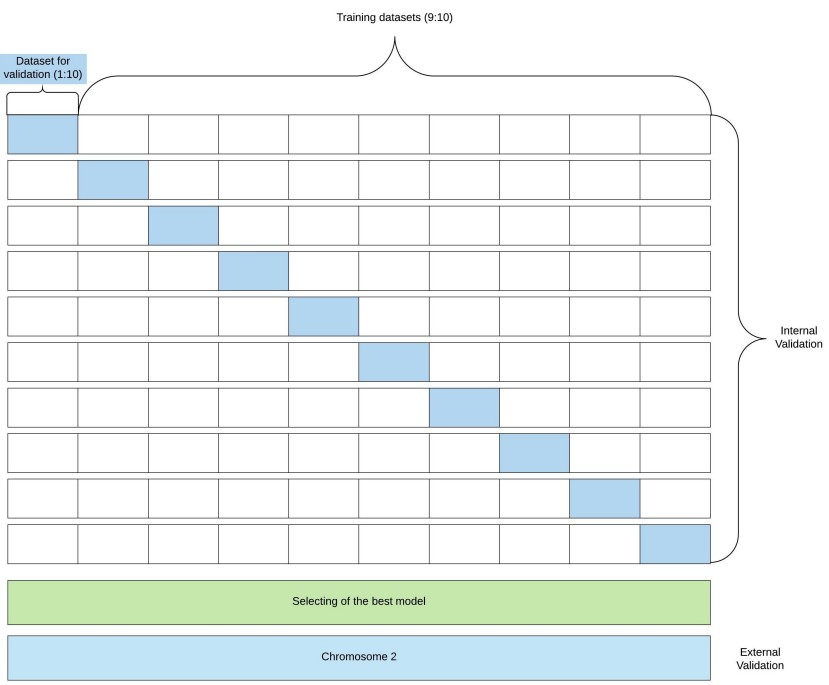

**Figure 2   Flowchart of model building and validation.** The figure presents a flowchart of the model building and validation.

## RESULTS

### Classification

To reduce the influence of how the data is split on the resulting testing statistics, we carried out a 10-fold cross-validation (CV) procedure for "internal" validation and supplemented it by the "external" validation of the rice chromosome 2 (excluded from the training set). We have randomly divided the dataset into 10 partitions and used each partition as the testing data while training the model on the remaining partitions. The ROC (Receiver Operating Characteristic) curve represents dependence of sensitivity on the specificity. It is a graph showing the performance of a classification model at all classification thresholds. AUC-ROC curves for classifier obtained in 10-fold "internal" cross-validation are presented in Fig. 3. Accuracy = 0.88, Se = 0.84, Sp = 0.92, CC = 0.79, AUC = 0.94.

Then we have selected the best model and evaluated its performance on rice chromosome 2 ("external" validation dataset) and compared it with TSSPlant. The dataset for "external" validation contained 2,000 nucleotide sequences with length 512 nt, with 1,000 examples— "non-TSS" sequences and 1,000—"TSS" sequences.  For the "external" validation we calculated the Matthews correlation coefficient (MCC), Accuracy (Ac), Sensitivity (Se), Specificity (Sp), and Area Under the ROC Curve (AUC-ROC) for classification models. The results of the "external" validation are presented in Table 1.

We have clearly shown that the classification model (Stage 1 of the pipeline) already achieves high accuracy (0.88). Regression (Stage 2) will further refine the prediction.

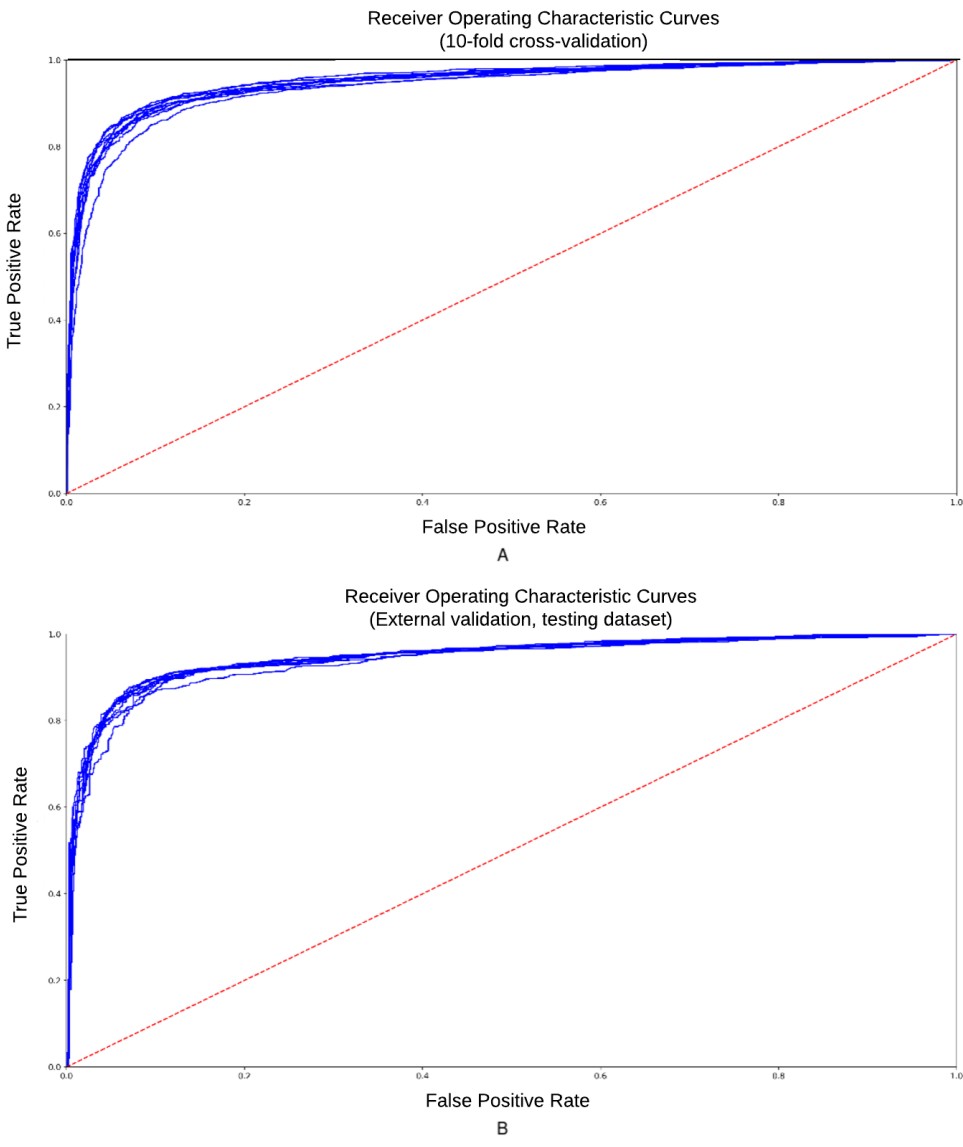

**Figure 3 ROC curve for the training and testing datasets.** ROC-curves obtained 10-fold cross-validation procedure of classification model. (A) Training dataset; (B) testing dataset. Accuracy = 0.88, Se = 0.84, Sp = 0.92, CC = 0.79, AUC = 0.94.

**Table 1 Comparison of accuracy metrics of TransPrise and TSSPlant classification models.** TransPrise offers higher accuracy compared to the TSSPlant.

| Classification model | MCC | Accuracy | Sensitivity | Specificity | AUC |
|---|---|---|---|---|---|
| TSSPlant | 0.310 | 0.603 | 0.976 | 0.231 | 0.603 |
| TransPrise | 0.791 | 0.895 | 0.872 | 0.919 | 0.952 |

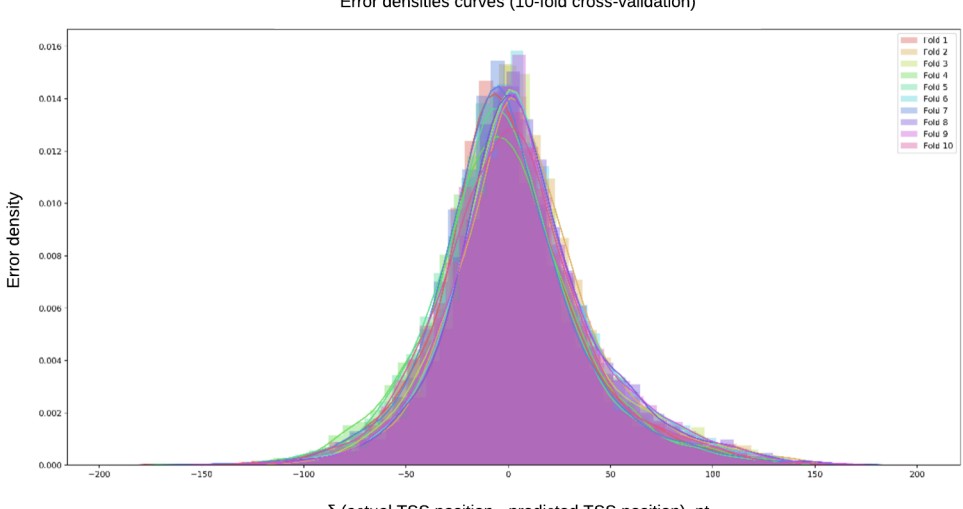

**Figure 4 Error density curves obtained in 10-fold cross-validation of regression models.** The error density curves were obtained in the 10-fold cross validation procedure for regression models. The mean absolute error (MAE) for regression model was 47.99 nt. The testing was performed for entire chromosomes, without filtering for possible TSS containing regions, located upstream of the ATG.

## Regression

Figure 4 shows the error density curves obtained in the 10-fold cross validation procedure for regression models. The mean absolute error (MAE) for regression model was 29.19 nt. The mean difference between predicted and true TSS, if the guess is random, is 66 nt.

We compared the accuracy of promoter prediction for TSSPlant and TransPrise. For this purpose, we selected rice chromosomes 1 and 2. The testing was performed for entire chromosomes, without filtering for possible TSS containing regions, located upstream of the ATG.

These two chromosomes contain 5,298 high quality experimentally validated TSS, defined as follows:

(1) Locus does not correspond to transposable element
(2) Locus has experimental support (full-length mRNA)
(3) If multiple isoforms are predicted, the "representative" is used
(4) Size of the 5′ UTR is at least 20 nt.

For these two chromosomes, TSSPlant predicted 153,009 TSSs, while TransPrise has found 13,765 sites. Of TSSPlant predictions, 10,721 (∼7%) were located within 1,000 nt from validated TSS. Of TransPrise predictions, 3,989 (∼29%) were located within 1,000 nt from validated TSS. Additionally, TransPrise predictions tend to be closer to validated TSS than TSSPlant predictions (Fig. 5).

## Validation of TransPrise in *Homo sapiens*

We have used Cap Analysis of Gene Expression (CAGE) data from the DBTSS database (https://dbtss.hgc.jp/), using the HG19 version of the human genome (*Suzuki et al., 2018*). The model was trained on all chromosomes except chromosome 8 and tested on
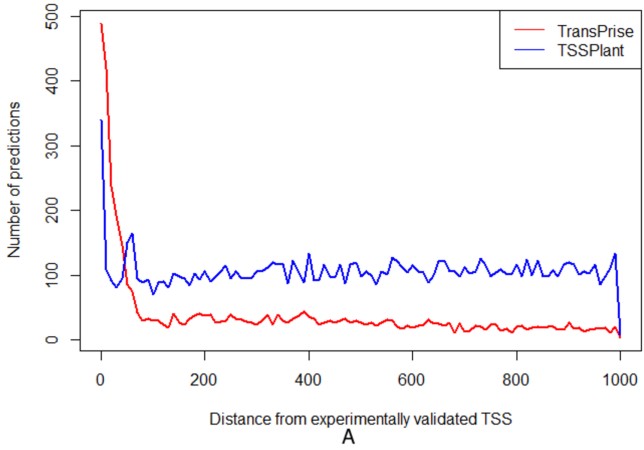

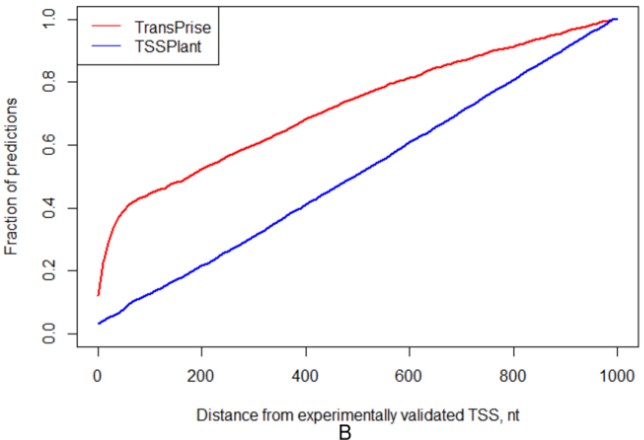

**Figure 5** **Comparison between TransPrise and TSSPlant predictions (fraction of sites).** TransPrise predictions tend to be closer to validated TSS than TSSPlant predictions. (A) Number of predictions; (B) fraction of prediction.

chromosome 8. Accuracy of the classification model was 0.778, sensitivity 0.816, specificity 0.74, and Matthews correlation coefficient 0.57. The mean absolute error of the regression model was 47.986 nt.

## DISCUSSION

We have developed an efficient deep learning approach for prediction of the position of transcription start sites in eukaryotes using properties of a nucleotide sequence. The approach is data-type independent and allows incorporation of additional data types (such as RNA-seq and tissue specific DNA methylation), refining positions of TSS for tissue-specific and stress-specific expression.

We compared TransPrise with the TSSPlant approach on an independent test set composed of 2,000 nucleotide sequences. All sequences were 512 nt long, and 1,000 sequences did not contain TSS ("non-TSS"), and 1,000 contained TSS ("TSS" sequences).

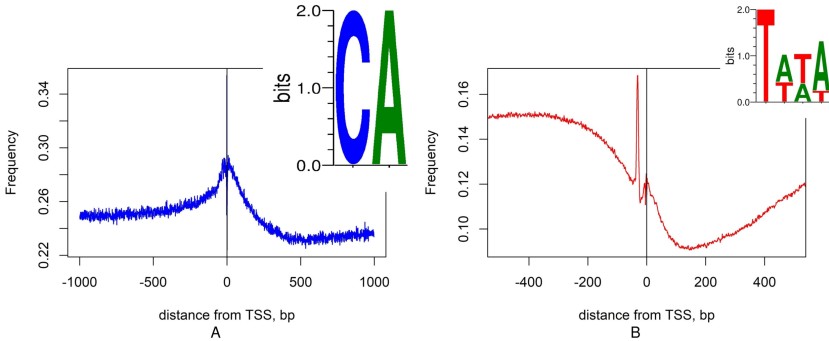

**Figure 6** **Distribution of CA and TATA motifs in promoter region.** (A) Distribution of the CA motif has a peak at TSS (position 0). (B) TATA motif is frequently located at position −30, its density shows a peak at −30.

The Matthews correlation coefficient value for TransPrise is more than twice larger than for TSSPlant classification models (0.79 vs. 0.31), indicating the significantly higher efficiency of TransPrise in distinguishing between regions that contain and do not contain starts of transcription. Additionally, a regression model was created for precise localization of TSS within the sequence classified as a "promoter". We validated our regression model on a test set composed of 1,000 "TSS" sequences selected from chromosome 2 and calculated the mean absolute error to be 47 nt.

Another important genome annotation task is identification of functional motifs. The architecture of TransPrise is especially designed for that. The first convolution layer is composed of four different kernel size filters ($a_{i,j}$)—$4 \times 2$, $4 \times 4$, $4 \times 8$, $4 \times 16$ matrices, where [$i$:[1,0,0,0](A),[0,1,0,0](T),[0,0,1,0](C),[0,0,0,1](G)] and [$j$:2,4,8,16—length of motif sequence] (in total 128 filters of each type). After model training, the filters correspond to PWM (position-specific weight matrix) describing informative sequences in promoters and can be visualized as sequence logos. Several filter motifs correspond to known regulatory elements: TGGGCC (*Lu et al., 2013*), CGATT (*Chen et al., 2016*), ACTCAT (*Weltmeier et al., 2006*), and CGCG box (*Yang & Poovaiah, 2002*). Motif TGGGCC is targeted by the TCP transcription factor through its interaction with proliferating cell nuclear antigens PCF1 and PCF2 (*Lu et al., 2013*); ACTCAT motif is a typical binding site of basic leucine zipper (bZIP) transcription factor (*Weltmeier et al., 2006*); CGCG cis-elements are found in promoters of stress-related genes, for example involved in ethylene signaling, abscisic acid signaling, and light signal perception. They are bound by AtSR1 transcription factor (*Yang & Poovaiah, 2002*).

Figure 6 shows "filter" motifs that correspond to two well-characterized features of eukaryotic promoters: Initiator element CA and TATA-box (*Smale & Baltimore, 1989*; *Zhu, Dabi & Lamb, 1995*). Therefore, we have shown that at least some of the features selected by the model as informative for identification of TSS correspond to known, biologically validated regulatory elements, over-represented at or near the start of transcription. We believe that other features may correspond to unknown regulatory elements.

## CONCLUSIONS

TransPrise is an advanced tool for improving genome annotation. This is symbolic of a mathematical approach to biology, which is increasingly significant. As the pressure to annotate genomes increases with plummeting sequencing costs, we must focus efforts on understanding the significance of individual genomic regions. TransPrise can predict important regulatory regions and identify characteristic motifs at a high level of efficiency relative to existing approaches. TransPrise is data-type and species independent tool that can be easily installed and customized.

### Funding

The authors received no funding for this work.

### Competing Interests

Tatiana Tatarinova is an Academic Editor for PeerJ. Dmitry Sokolov is employed by Neirika Solutions and that Duane R. Chartier is the president and CEO of the International Center for Art Intelligence, Inc.

### Author Contributions

- Stepan Pachganov and Aleksei Zarubin performed the experiments, analyzed the data, prepared figures and/or tables, approved the final draft.
- Khalimat Murtazalieva analyzed the data, prepared figures and/or tables, approved the final draft.
- Dmitry Sokolov contributed reagents/materials/analysis tools, prepared figures and/or tables, approved the final draft, website development, designed the software.
- Duane R. Chartier analyzed the data, authored or reviewed drafts of the paper, approved the final draft.
- Tatiana V. Tatarinova conceived and designed the experiments, analyzed the data, prepared figures and/or tables, authored or reviewed drafts of the paper, approved the final draft.

### Data Availability

The code is available at http://www.compubioverne.group/data-and-software/.

Data is available at GitHub: https://github.com/StepanAbstro/TransPrise.

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
