# Peer review of "TransPrise: a novel machine learning approach for eukaryotic promoter prediction"

_PeerJ, doi:10.7717/peerj.7990_

## Round 0.1 · original submission · Major Revisions

The manuscript received critical comments from two reviewers. The remarks are rather technical. Please check correspondence between Table 1 and other captions.

Reviewer 1 ·

Basic reporting

The authors of this well-written, easy-to-read, clearly illustrated, and unambiguously understandable manuscript are proposing their new machine learning Web-service TransPrize for recognizing transcription start sites (TSS) upon a whole genome DNA sequences via demonstrating its workability on the example of the RNAseq-data on the rice genome (except chromosome 2 as a control dataset) in comparison with the well-known Web-service TSSPlant. Since the prediction accuracy expressed within a number of statistics , which was achieved by the authors seems to be significantly than those of the commonly accepted tools under comparison, it would be interesting for many readers of PeerJ, I hope. In my opinion, both main sections - Background and References, - are sufficiently brief and adequate to the current state of this field of intense research without minor details. As I understood from the manuscript, the only public data were studied here.

Experimental design

Since the results of the calculations were reproduced by the authors on 10 variants of independent data, as well as the contextual patterns, which they identified, are cjrresponding the most frequent regulatory signals known, such as INR-element, TATA-box, and several transcription factor binding sites, then the originality of these results and their reliability are beyond doubt. In my opinion, the description of all the technical details seems to be sufficient to replicate them.

Validity of the findings

Since the authors present the results of their Web service TransPrize on the only example of the rice chromosome-2, then, in my opinion, their manuscript could look much more better and mature, if all the phrases presenting here about an applicability of this tool to analyze any other eukaryotic species and their viruses would have been mitigated to subjunctive assumptions.

Additional comments

As one minor comment, the statistical characteristics of the predictions made using TransPrize, which are shown in Table 1, do not coincide with those in both main text and caption for Figure 3 so that this mismatch should be either corrected or discussed.

Taking the all above mentioned, I am recommending this manuscript for publication as soon as the above mismatch will have been eliminated.

Reviewer 2 ·

Basic reporting

Prediction of transcription start sites (TSS) is considered a difficult problem, because of lack of conserved elements around TSS and also possibility of multiple alternative positions per single gene. Recent progress in PacBio’s Isoseq technology for sequencing of full-length transcripts partially alleviated the need for computational predictions, as TSS could be found by direct mapping of full-length transcripts to the genome. Nonetheless, prediction of TSS based on sequence alone, as attempted by authors, presents some interest, e.g. for genomes without full-length transcriptome data.
However current manuscript needs some major revisions. Below are some suggestions:
1. Abstract should include some information about accuracy of method
2. Deep convnets usually have a lot of parameters, it is not clear from the text if enough number of sequences were used for training of parameters, fewer data may lead to overfitting
3. As it is a 2-stage algorithm it would be helpful to provide some probability/score measure about reliability of ‘promoter’ prediction after 1st stage.
4. Authors report average error shift for exact TSS prediction at about ~48nt in 512nt length window. It is not clear how good it is relative to e.g, random predictions.
5.Comparison with TSSPlant was performed on chr.1 and 2 of O.sativa, the species where algorithm was trained, even if earlier authors indicate that chr2 was used for validation. If this method intents to be not species-specific, it is better to test in another species, preferably where full-length transcriptome data available.

Some minor problems:
1.it is difficult to read manuscript in PFD format, as some figures have nonstandard sizes
2. line 94: ‘did use unreliable homology arguments..’ - not clear what means by that phrase.
3.line: 194: ‘4 kernel sizes’ but at line 197: ‘There were one kernel size and ReLU in both situations’

Experimental design

no comment

Validity of the findings

no comment

---

## Round 0.2 · accepted · Accept

Both reviewers recommended to accept the paper. Please fix the typos (abbreviations in the figure legends) noted by the second review. I believe it doesn't demand new reviewing round.

Reviewer 1 ·

Basic reporting

Since the authors successfully satisfied all my comments in the revised manuscript, I recommend accepting it for publication.

Experimental design

Since the authors successfully satisfied all my comments in the revised manuscript, I recommend accepting it for publication.

Validity of the findings

Since the authors successfully satisfied all my comments in the revised manuscript, I recommend accepting it for publication.

Additional comments

Since the authors successfully satisfied all my comments in the revised manuscript, I recommend accepting it for publication.

Reviewer 2 ·

Basic reporting

Some minor points in text to change:

line 138: 2 times 'deep learning'
line 186: 512nt
Fig1. Meaning of abbreviations BN and MP in figure (batch normalizations and max.pooling)
Fig3 ROC curve for the training (and testing) datasets
Fig4. should mention that data for humans and distance is 47.99 (not 47.68 - typo)

Experimental design

Methods described with sufficient detail and software is available

Validity of the findings

Results are valid

Additional comments

Authors revised the paper, according to reviewers suggestions, and I think in current form it is publishable in the journal.